# Exploring Core Microbiota Based on Characteristic Flavor Compounds in Different Fermentation Phases of Sufu

**DOI:** 10.3390/molecules27154933

**Published:** 2022-08-03

**Authors:** Wei Wu, Zhuochen Wang, Boyang Xu, Jing Cai, Jianghua Cheng, Dongdong Mu, Xuefeng Wu, Xingjiang Li

**Affiliations:** 1School of Food and Biological Engineering, Hefei University of Technology, Hefei 230009, China; 2020171476@mail.hfut.edu.cn (W.W.); xuboyang0904@163.com (B.X.); caijing@hfut.edu.cn (J.C.); d.mu@hfut.edu.cn (D.M.); wuxuefeng@hfut.edu.cn (X.W.); 2Institution of Agricultural Products Processing, Anhui Academy of Agricultural Sciences, Hefei 230001, China; wangzhuochen123@hotmail.com (Z.W.); chengjianghua123@hotmail.com (J.C.)

**Keywords:** sufu, bacterial community, flavor components, co-occurring network, core microbiota

## Abstract

Sufu, a Chinese traditional fermented soybean product, has a characteristic foul smell but a pleasant taste. We determined the core functional microbiota and their metabolic mechanisms during sufu fermentation by examining relationships among bacteria, characteristic flavor compounds, and physicochemical factors. Flavor compounds in sufu were detected through headspace solid-phase microextraction coupled with gas chromatography–mass spectrometry, and the microbial community structure was determined through high-throughput 16S rRNA sequencing. The results showed that the fermentation process of sufu could be divided into early and late stages. The early stage was critical for flavor development. Seven microbiota were screened based on their abundance, microbial relevance, and flavor production capacity. Five microbes were screened in the early stage: *Pseudomonas*, *Tetragenococcus*, *Lysinibacillus, Pantoea,* and *Burkholderia–Caballeronia–Paraburkholderia*. Three microbes were screened in the late stage: *Exiguobacterium*, *Bacillus*, and *Pseudomonas*. Their metabolic profiles were predicted. The results provided a reference for the selection of enriched bacterial genera in the fermentation process and controlling applicable process conditions to improve the flavor of sufu.

## 1. Introduction

Sufu, a Chinese traditional fermented soybean food with a characteristic unique flavor, rich mellow taste, and high nutritive value, is also known as Chinese cheese because of its similarity to cheese in shape, texture, and fermentation mechanisms [1]. Sufu has high nutritional value, being rich in B vitamins and protein; due to microbial fermentation, protein is converted into various free amino acids, so it has the effect of promoting the appetite and digestion. Sufu can be divided into four categories based on the type of starter culture used: bacteria fermented (inoculated with *Bacillus* or *Micrococcus*), mold-fermented (inoculated with *Mucor*, *Rhizopus*, or *Actinomucor*), naturally fermented (naturally inoculated), and enzyme ripened (added with protease preparations). Among them, mold-fermented sufu is the most common. Four steps are involved in the production of mold-fermented sufu: (i) preparation of tofu cubes (soybeans are ground and mixed with water in a ratio of 1:6, solidified, pressed into shape, and cut into pieces), (ii) pre-fermentation (inoculating tofu pieces with *Mucor* to prepare pehtzes), (iii) salting (salting of pehtzes), and (iv) post-fermentation (aging them for approximately three months in bottles containing the dressing mixture) [2].

Among these steps, post-fermentation is a natural process in which multiple species work together to produce various enzymes, such as proteases and lipases. These enzymes hydrolyze proteins and fats into amino acids, fatty acids, and small molecule peptides, which act as precursors to flavor compounds. Although multiple bacterial strains are involved in the post-fermentation process, only a few key microorganisms were reported to drive the fermentation process [3]. Xie et al. examined the correlation between the microbiota and characteristic flavor compounds in different types of sufu and observed that *Lactococcus*, *Sphingobacterium*, *Pichia*, *Kodamaea*, and *Saccharomyces* considerably contributed to the development of flavor components [1]. He et al. evaluated the correlation among microbiota, flavor compounds, and physicochemical parameters and identified nine bacteria (*Bacillus*, *Tetragenococcus*, *Enterobacter*, *Lactobacillus*, *Stenotrophomonas*, *Sphingobacterium*, *Trabulsiella*, *unclassified*, and *Weissella*) and six fungi (*Alternaria*, *Sterigmatomyces*, *Debaryomyces*, *Fusarium*, *Candida*, and *Actinomucor*) that were core microbiota and played vital roles during sufu fermentation [4]. Huang et al. reported that the flavor and taste of sufu were mainly affected by *Enterobacter* and *Lactococcus* [5]. However, these studies only focused on the abundance of genera and their ability to produce flavor compounds and did not investigate the effect of interactions between genera on flavor compounds. Previous studies indicated that some microbes are not effective producers of flavor compounds during fermentation; however, they indirectly induce the production of flavor compounds by facilitating the growth of flavor compound producers [6,7]. In addition, complex interactions between microbes maintain the stability of microbial networks. Therefore, microbial interactions can considerably affect the formation of flavor compounds.

In this study, to elucidate mechanisms involved in the post-fermentation process of sufu, we divided the whole post-fermentation process into two stages. First, we screened the core microbiota involved in the fermentation process by performing flavor-generation and co-occurrence network analyses and determined their metabolic pathways using Kyoto Encyclopedia of Genes and Genomes (KEGG). Second, we identified key physicochemical factors affecting core microorganisms.

## 2. Results

### 2.1. Physicochemical Parameters of Sufu during Post-Fermentation

Appendix A lists the total acidity and salt, moisture, and ethanol contents determined during the production of sufu. The moisture content remained stable during the ripening period. The ethanol content fluctuated in the range of 12.67 to 20.24 mg/g. the source of the ethanol was from the high concentration of baijiu added to sufu and microbial metabolism during production [8]. Total acidity was a crucial indicator of the maturity of sufu and was helpful for sufu preservation [9]. Total acidity increased between day 0 and day 90 of fermentation and then decreased slightly until the end of ripening. Similarly, the salt content was among the most crucial factors affecting the quality of sufu; it was the highest at day 0, increased from day 10 to day 30, and held steady between day 60 and day 130.

### 2.2. Dynamics of Flavor Compounds during Sufu Fermentation

A total of 69 flavor compounds were detected in the sufu samples. Compounds with an OAV of ≥1 are regarded as characteristic volatile flavor compounds [10]. As shown in Table 1, 41 volatile flavor compounds with an OAV of ≥1 included 14 esters, 8 alcohols, 11 aldehydes, 3 ketones, 2 phenols, 1 acid, 1 furan, and 1 pyrazine. Twelve of these compounds (ethyl propionate, ethyl 2-methyl butyrate, ethyl hexanoate, ethyl oenanthate, ethyl caprylate, ethyl isobutyrate, 1-octene-3-ol, linalool, benzaldehyde, phenylacetaldehyde, eugenol, and 2-pentylfuran) were reported to considerably contribute to sufu flavor [1,11]. Other compounds with desirable aroma were found in many bean products, including ethyl butanoate, ethyl caprate, 2-octyne-1-ol, (E)-2-nonenal, nonanal and (E, E)-2, 4-octandienal [12,13,14,15]. Furthermore, 2,5-dimethyl-3-ethylpyrazine with a roast smell was first discovered as a characteristic flavor compound during sufu fermentation. Esters provide a fruit-like odor and improve the flavor of sufu. Ethyl propionate (pineapple-like odor), ethyl 2-methyl butyrate (grassy odor), ethyl hexanoate (almond- and apple-like odors), ethyl oenanthate (fruity odor), ethyl caprylate (creamy odor), and ethyl isobutyrate (nail polish odor) were detected in other fermented soybean products [16,17]. 1-Octene-3-ol, which imparts a mushroom-like odor to sufu, is an enzymatic product of fatty acids and considerably contributes to the flavor of fermented soybean products [18,19]. Benzaldehyde (honey odor) and phenylacetaldehyde (orange-like odor) were regarded as characteristic aroma components and flavor enhancers in fermented soybean products due to their low thresholds. 2-Pentylfuran with a beany smell was also detected in fermented products, and might be a product of the natural oxidation of linolenic acid, which provides an undesirable flavor to sufu [20].

PCA was performed to examine differences in volatile compounds among the sufu samples. As presented in Appendix A, the PCA biplot indicated that the first and second principal components explained 53.8% and 19.7% of the total variation, respectively, during day 0 to day 30. The long distances among the four sample points indicated high variation in flavor composition, and the points were distributed in the second, third, and fourth quadrants, which were mainly characterized by 2,4-undecadienal, decanal, (E, E)-2, 4-octandienal, phenylacetaldehyde, octanal, hexyl alcohol, linalool, and 2,5-dimethyl-3-ethylpyrazine. Sufu samples from day 60 to day 130 were grouped and located in the first quadrant characterized by ethyl propionate, ethyl caprylate, ethyl hexanoate, ethyl oenanthate, pentanal and hexanal. Most of the characteristic aldehydes and alcohols were detected during day 0 to day 30. Moreover, some characteristic esters were observed during day 60 to day 130; this finding is in keeping with that reported by [14]. In conclusion, the stage from day 0 to day 30 is crucial for flavor formation.

Six characteristic flavors of sufu were selected: sour, floral, fruity, alcoholic, grease, and fermented. As shown in Figure 1, from day 0 to day 130 of fermentation, the intensity of floral flavor of sufu was low, fluctuating in the range of 0–0.5, the intensity of sour flavor fluctuated between 0–1, and the fruit flavor reached a maximum value of 2.5 on day 90, which might be related to the accumulation of various esters during late stage, such as ethyl butanoate, ethyl isobutyrate, ethyl valerate, ethyl oenanthate, and 4-decanolide(Table 1). The alcoholic flavor was strongest on day 10, which might be attributed to the high concentration of alcohols in the early stage, such as thanol and isoamyl alcohol. The fermented flavor became stronger as the fermentation progresses. The grease flavor was higher in intensity in the early stage compared to the late stage, reached its maximum on day 20. This might be due to the fact that the characteristic aldehydes contributing to the grease flavor accumulate mainly in the early stage.

### 2.3. Dynamics of Microorganisms during Sufu Post-Fermentation

#### 2.3.1. Alpha Diversity during the Fermentation Process

Due to the low abundance of fungi in the samples, the results of fungal PCR amplification did not meet the conditions for library building on the machine. Thus, only 16S rDNA high-throughput sequencing was performed on these samples. A total of 802,520 optimized sequences were obtained from the 21 sufu samples with an average sequence length of 428 bp. The coverage rate of high-quality sequences was >99%, indicating the reliability of sequencing results.

The alpha diversity indices, namely Chao1, ACE, Shannon, and Simpson, for bacteria, are presented in Figure 2. Chao1 and ACE represented community richness, whereas Shannon and Simpson reflected community diversity. The abundance and diversity of bacteria tended to decrease from day 0 to day 10, possibly due to the bacteriostatic effect of condiments present in the dressing mixture [4]. Subsequently, the abundance and diversity of bacteria rapidly increased and reached a maximum at day 30 and stabilized in the bacterial community between day 60 and day 130; these findings are similar to those reported by [4,5]. On the basis of the variation of alpha diversity, the post-fermentation of sufu could be divided into two stages: early stage (days 10–30) and late stage (days 60–130).

#### 2.3.2. Composition of Bacterial Communities

We analyzed the bacterial community composition at both the early and late stages (Figure 3). We identified 11 dominant genera (relative abundance > 1%) in the early stage and 19 dominant genera in the late stage. As presented in Figure 3a, *Bacillus*, *Acinetobacter*, and *Lysinibacillus* were predominant at day 10. In particular, the genus *Bacillus* accounted for approximately 80% with the progression of fermentation. The abundance of *Bacillus* gradually decreased until it became a minority population at the end of ripening, and Lysinibacillus followed the same trend as *Bacillus*. *Burkholderia–Caballeronia–Paraburkholderia* significantly increased during the early stage and became predominant at day 30 (25.23%). As presented in Figure 3b, the species of the dominant genera in the late stage increased significantly, and the abundance of these dominant genera tended to be more homogeneous in the late stage compared with the early stage. *Bacillus*, *Lysinibacillus*, and *Staphylococcus* gradually disappeared in the late stage. Other genera including *Burkholderia–Caballeronia–Paraburkholderia*, *Macrococcus*, *Pseudomonas*, and *Exiguobacterium* grew and rapidly multiplied to become the dominant microbes (Appendix A).

The transformation of the dominant genera from the early stage to late stage might be attributed to changes in environmental conditions, including higher acidity and temperature and lower oxygen content in the late stage, which inhibited the growth of microorganisms intolerant to the harsh environment required for sufu fermentation [15]. Thus, some acid-tolerant bacteria, such as *Tetragenococcus* and *Pseudomonas*, increased rapidly in the late stage. However, highly resistant *Bacillus* and *Lysinibacillus* tended to decrease throughout the fermentation stage and became a minority genus (relative abundance <1%) at 130 d, possibly due to competition between bacterial genera [22].

### 2.4. Correlation Analysis of Dominant Genera and Characteristic Flavor Compounds

In the early stage, we calculated the Spearman correlation coefficient between the 11 dominant genera and 40 characteristic flavor compounds. Bacterial genera with |r| > 0.7 and *p* < 0.05 and high correlations with more than 12 flavor compounds were considered to be strongly associated with flavor [15]. As presented in Figure 4, seven bacterial genera, namely *Pseudomonas*, *Tetragenococcus*, *Lysinibacillus*, *Bacillus*, *Pantoea*, *Staphylococcus*, and *Burkholderia–Caballeronia–Paraburkholderia*, were strongly correlated with 32 characteristic flavor compounds. Among them, *Bacillus* and *Lysinibacillus* were negatively correlated with characteristic flavor compounds. In contrast, the other five genera were positively correlated with characteristic flavor compounds and promoted flavor formation.

We calculated the Spearman correlation coefficient between 19 dominant genera and 41 characteristic flavor compounds. Genera with |r| > 0.7 and *p* < 0.05 and high correlations with more than seven flavor compounds were considered to be strongly associated with flavor. As presented in Appendix A, the contribution of microbiota to flavor compounds decreased in the late stage than in the early stage. We screened four genera: *Exiguobacterium*, *Pseudomonas*, *Lactococcus*, and *Bacillus*. *Pseudomonas* and *Bacillus* continued to function in the later stage, indicating that they could adapt to the fermentation environment. *Bacillus* was negatively correlated with ethyl isovalerate and linalool and positively with hexanal, heptanal, octanal, and nonanal. *Exiguobacterium* was positively correlated with ethyl isovalerate, ethyl caprate, ethyl benzoate, and linalool. *Pseudomonas* was positively correlated with ethyl isovalerate and linalool.

### 2.5. Co-occurrence Network Analysis during Sufu Fermentation

We performed a co-occurrence network analysis to examine interactions between microorganisms [6]. We calculated the Spearman correlation coefficient of 11 dominant genera in the early stage. The genera with |r| > 0.7 and *p* < 0.05 were considered to be significantly correlated, and node size represents the magnitude of its degree, which indicates the number of the edges of a node; a larger degree indicates a more crucial role of the node in the co-occurrence network. As presented in Figure 5a, *Lysinibacillus* and *Bacillus* exhibited an antagonistic relationship with other four flavor-producing bacteria, namely *Pseudomonas*, *Tetragenococcus*, *Pantoea*, and *Burkholderia–Caballeronia–Paraburkholderia*, and these four genera exhibited a synergistic relationship with each other. *Lysinibacillus* and *Bacillus* inhibit the growth of other microorganisms by generating various metabolic components [23,24]. These results are consistent with those of the analysis of correlations between dominant genera and characteristic flavor compounds.

A co-occurrence network analysis of the 19 dominant genera in the late stage was performed and significant connections of correlation coefficients |r| > 0.7 and *p* < 0.05 were visualized in Figure 5b. On the basis of its central location in the co-occurrence network, *Bacillus* was determined to play a significant role in the late stage. *Bacillus* was negatively correlated with other flavor-producing bacteria. Significant positive correlations were noted among *Exiguobacterium*, *Corynebacterium*, *Tetragenococcus*, *Pseudomonas*, and *Aerococcus*, indicating that *Bacillus* exhibited an antagonistic relationship with other dominant genera throughout the fermentation stage.

### 2.6. Identification of Core Microbiota during Sufu Fermentation

Based on three criteria, (1) dominant bacteria with >1% abundance; (2) strongly flavor correlated bacteria; (3) the genera with degree >4 in the co-occurrence network [6,15,25], *Pseudomonas*, *Tetragenococcus*, *Lysinibacillus*, *Pantoea*, and *Burkholderia–Caballeronia–Paraburkholderia* were determined as core microbiota in the early stage (days 10–30) and *Exiguobacterium*, *Bacillus*, and *Pseudomonas* in the late stage (days 60–130). Among these, *Tetragenococcus* and *Bacillus* were identified as the core functional microbiota in plain sufu [4,15]. In previous studies, *Lactococcus* was repeatedly reported to make great contributions to sufu fermentation, *Sphingobacterium* and *Enterobacter* were also proven to be the key microorganisms in sufu products [4,26]. although they were dominant during sufu fermentation (Figure 3); however, no correlation was observed between *Sphingobacterium* and *Enterobacter* and volatile compounds in our studies.

*Bacillus* inhibited the formation of most characteristic flavor compounds in the early stage and promoted the production of some short-chain fatty aldehydes (hexanal, heptanal, nonanal, and (E)-2-nonenal) in the late stage. All these compounds were produced through lipid oxidation or degradation. Hexanal contributed an unpleasant grass-like aroma to sufu. Heptanal, nonanal, and (E)-2-nonenal contributed a fatty-like odor to sufu. *Bacillus* exerted a detrimental effect on sufu flavor; this finding is consistent with that reported by [5,27], who demonstrated that *Bacillus* inhibits flavor production during sufu fermentation. However, *Bacillus* is often used to fortify Daqu during the production of white wine, and the flavor of Daqu enriched with *Bacillus* is better than the conventional Daqu [27]. Similar to *Bacillus*, *Lysinibacillus*, which is rarely reported, inhibited the production of flavor compounds.

*Tetragenococcus* promoted the formation of aroma compounds in the early stage and was identified as a crucial contributor to flavor formation in fermented soy products [15,16,28]. In contrast to the more abundant *Bacillus*, the less abundant *Tetragenococcus* appeared to be the initiator of the fermentation process in soy products [29].

*Pseudomonas* promoted the formation of flavor compounds in the early stage. A study reported that *Pseudomonas* exerts a strong effect on flavor production in other fermented foods [30]. Moreover, *Pseudomonas* can degrade fats and proteins to produce flavor precursors [31]. However, a negative correlation was observed between *Pseudomonas* and some fatty aldehydes in the late stage. Moreover, *Pseudomonas* was reported to be negatively correlated with flavor compounds and was often considered to be food-spoiling bacteria [32].

The genus *Exiguobacterium* emerged as the dominant genera and facilitated the formation of specific flavor compounds (ethyl caprate, ethyl benzoate, ethyl isovalerate, and linalool) by generating proteases and lipases in the late stage; this might be ascribed to its ability to grow and work at a broad range of pH and temperature [33]. The genus *Pantoea* is present in other fermented foods [34,35,36]. Zhao et al. reported that *Pantoea* promotes the formation of aroma compounds during glutinous rice wine fermentation, indicating that *Pantoea* is a major flavor producer. *Burkholderia*–*Caballeronia*–*Paraburkholderia* was found to be predominant in doubanjiang and pit mud for baijiu production [37,38]. Few studies reported on the relationship between *Burkholderia*–*Caballeronia*–*Paraburkholderia* and flavor compounds. *Burkholderia*–*Caballeronia*–*Paraburkholderia* exhibited positive correlations with flavor compounds probably due to its ability to produce lipases, which decompose lipids and generate volatile flavor compounds.

### 2.7. Correlation Analysis of Core Microbiota and Physicochemical Factors

Environmental factors, namely the salt, moisture, and alcohol contents and acidity, affect the bacterial community structure during sufu fermentation. We performed a redundancy analysis (RDA) to determine the contributions of these environmental factors. As presented in Figure 6a, the findings of RDA analysis indicated that the environmental factors accounted for 73.39% of the variation in the bacterial community in the early stage. Total titratable acidity and salt and ethanol contents exerted a more pronounced effect on the core microbiota compared with the moisture content. Total titratable acidity and the salt content were positively correlated with *Tetragenococcus*, *Burkholderia*–*Caballeronia*–*Paraburkholderia*, *Pantoea*, and *Pseudomonas* (Figure 6b); this finding might be attributed to their ability to tolerate elevated levels of acid and salt [39]. The ethanol content was positively correlated with *Lysinibacillus* and negatively correlated with the other bacteria. Ethanol exerts an inhibitory effect on the activity of microbially induced proteases, thus reducing the degradation of proteins and the generation of flavor precursors and inhibiting the production of flavor compounds [40].

As presented in Figure 6c, the results of canonical correspondence analysis indicated that the four physicochemical factors explained 46.3% of the variance in the bacterial community, indicating that the contribution of these factors to the overall bacterial community decreased with extended fermentation. The dominant bacteria in the late stage more favorably adapted to changes in the environmental factors compared with those in the early stage (Figure 6d). *Bacillus* was negatively correlated with the ethanol content and positively with the salt content, indicating that *Bacillus* is tolerant to salt but not ethanol. In addition, we observed that *Pseudomonas* and *Exiguobacterium* were positively correlated with ethanol; they could produce various hydrolytic enzymes that degraded sufu substrates to produce secondary metabolites including alcohols [31,41].

### 2.8. Metabolic Pathways of the Core Microbiota during Sufu Fermentation

We performed KEGG analysis based on PICRUSt2 to determine the metabolic pathways of the seven core genera throughout fermentation. As presented in Figure 7, cellulose and starch were degraded into glucose by cellulase and amylase, respectively. *Bacillus*, *Pantoea*, *Exiguobacterium*, and *Lysinibacillus* were responsible for saccharification. Subsequently, glucose was converted into pyruvate, which was the key intermediate metabolite involved in lactate and alcohol fermentation. *Tetragenococcus* was the main microorganism promoting the formation of lactate and ethanol [17]. Proteins were broken down by proteases into various free amino acids mainly by *Bacillus*, *Exiguobacterium*, *Pseudomonas*, and *Lysinibacillus*. Some aromatic amino acids, such as leucine and isoleucine, were degraded into 3-methylbutanal and 2-methylbutanal, respectively, and then converted into isoamyl alcohol, isovaleric acid, and 2-methylbutanoic acid mainly by *Tetragenococcus*, *Exiguobacterium*, and *Pantoea*. Lipids were broken down by *Bacillus*, *Burkholderia*–*Caballeronia*–*Paraburkholderia*, and *Pseudomonas* into fatty acids, among which short-chain fatty acids, such as octanoic acid and heptanoic acid, were converted into corresponding fatty aldehydes and fatty alcohols. Esters were produced by the enzymatic reactions of acids and alcohols. *Pantoea* and *Exiguobacterium* were involved in the synthesis of esterases [36] and promoted the production of esters in the early and late stages, respectively.

*Bacillus* can release various hydrolases including lipases, amylases, and proteases, and they promote the hydrolysis of lipids, starch, and protein into various flavor compounds [27]. The correlation analysis indicated that *Bacillus* had a negative relationship with flavor compounds in the early stage (Figure 4), and this could be attributed to the fact that Bacillus is involved in the decarboxylation of amino acids to produce biogenic amines by producing decarboxylase [17], which consumed substrates involved in amino acid metabolism, thus reducing the formation of aroma compounds. Biogenic amines are toxic substances commonly present in high-protein fermented foods, such as tempeh, natto, and sufu [42]. In addition, *Pseudomonas* is the major producers of biogenic amines [43]. With the significantly increased abundance of *Pseudomonas*, the production of some fatty aldehydes was inhibited by amino acid decarboxylation. Therefore, the abundance of *Bacillus* and *Pseudomonas* should be maintained at low levels to promote the production of aroma compounds during sufu fermentation.

## 3. Materials and Methods

### 3.1. Sample Collection

The sufu samples were obtained from a bean products factory in Bagongshan (Huainan, China). Seven independent batches of sufu samples were collected during post-fermentation on days 0, 10, 20, 30, 60, 90, and 130, namely d0, d10, d20, d30, d60, d90, d130. For each sample point, three parallel samples from different positions were used to analyze. These samples were passed back to the laboratory and stored at −80 °C for further experiments.

### 3.2. Physicochemical Property Determination

The sufu cubes were placed on a funnel for 30 min to remove the brine broth. Then, 0.5 cm of the skin was cut off the surface of the sufu pieces. The pieces were ground and mixed for physicochemical tests. The moisture content was determined by drying the samples to achieve a constant weight at 105 °C. Total acidity was measured by titrating with 0.01 M NaOH until the pH reached 8.2. The salt content was determined by titrating with 0.1 M silver nitrate standard solution until the solution was brick red. The ethanol content was analyzed through gas chromatography (Agilent Technologies, Palo Alto, CA, USA).

Volatile flavor compounds were analyzed using GCMS-QP2010 (Shimadzu, Japan) as described previously [44] with some modifications. Briefly, 2 g of the treated sufu sample, 4 mL of 0.25 g/mL brine, and 10 μL of 8.19 mg/L octanol (internal standard) were added to a sample vial, mixed, and equilibrated in a constant stirring water bath at 55 °C for 30 min. The 50/30-µm divinylbenzene/carboxen/polydimethylsiloxane fiber (Supelco, Inc., Bellefonte, PA, USA) was used to extract volatile flavor compounds for 30 min at 55 °C. The fiber was then inserted into a GC-MS inlet and desorbed for 3 min. The following procedure was followed to increase the temperature: the oven temperature maintained at 35 °C for 5 min, increased to 200 °C at a rate of 4 °C/min, and then held for 30 min. The injection temperature was set to 270 °C at a splitless mode. A solvent delay of 3 min was used. The ion source and interface temperature was set at 250 °C, and electron impact ionization was set at 70 eV in the range of 33–550 *m*/*z*.

The retention index (RI) of *n*-alkanes (C7–C40) under the same GC-MS conditions as the sample was used to calculate the retention index of each detected volatile flavor substance, compared it with the reference values given by the NIST database (mass-spectral similarity match ≥80), the formula for calculating the retention index was as follows:(1)$$\mathrm{RI}=100n+100\times \frac{t_{i}-t_{n}}{t_{n\;+1}-t_{n}}$$
where n and n + 1 represented number of carbon atoms of *n*-alkanes before and after the compound, t_n_ and t_n + 1_ represented retention time of *n*-alkanes, ti indicated Retention time of the compound (t_n_ < t_i_ < t_n + 1_).

The concentration of each compound was calculated based on the content and peak area of the internal standard as follows:(2)$$C=\frac{A_{x}\times C_{0}\times V}{A_{0}\times m}$$
where C was the flavor substance content (ug/100 g), A_x_ was the peak area of the flavor substance, A_0_ represented the peak area of internal standard, C_0_ represented concentration of internal standard, V was the injection amount of internal standard (uL), m was the mass of the sufu sample (g).

The odor activity value (OAV) was conducted to assess the contribution of each component to sufu and the OAV was calculated as follows eq:(3)$$\mathrm{OAV}=\frac{C}{T\times 100}$$
where C was the volatile flavor component concentration(ug/100 g), T was the detection threshold of volatile flavor component(ug/g).

### 3.3. DNA Extraction, Polymerase Chain Reaction, and Illumina MiSeq Sequencing Analysis

Polymerase chain reaction (PCR) of the samples was performed in Shanghai Majorbio Biomedical Technology Co., Ltd (Shanghai, China). using the Fast DNA SPIN extraction kit to extract Genomic DNA in sufu samples keeping to the manufacturer’s instructions. The V3-V4 regions of 16S rRNA genes were amplified using the universal primers 338F (5′-ACTCCTACGGGAGGGAGGA-3′) and 806R (5′-GGACTACHVGGGTWTCTAAT-3′). PCR was performed in triplicate in a 25 µL reaction mixture containing 2.5 µL of 10× Pyrobest reaction buffer, 1 µL of each primer, and 2 µL of dNTPs. The amplification program for PCR included an original denaturation procedure at 95 °C for 2 min, 35 amplification cycles (each cycle consisting of 95 °C for 30 s, 55 °C for 1 min, and 72 °C for 30 s), and a last incubation procedure at 72 °C for 10 min. Amplicons were identified through 2% agarose gel electrophoresis and quantified using QuantiFluor-ST (Promega, Beijing, China). The sequences of amplified products were analyzed using the Illumina Miseq sequencing platform.

### 3.4. Sensory Evaluation

The sensory evaluation of sufu followed the requirements of ISO 8586-2012. Fourteen evaluators were initially selected, the evaluators were screened to 10 through stimulus perception experiments, and they were trained to differentiate stimulus intensity levels; in brief, the six flavors (sour, floral, fruity, alcoholic, grease, and fermented) corresponding to the standards were phenylacetaldehyde, citric acid, isoamyl acetate, 4-vinyl-2-methoxyphenol, (E,E)-2,4-nonadienal, and ethyl caproate, were diluted in ethanol at four different concentrations and randomly presented to the evaluators, who are required to put them in order of increasing intensity. After the training, all evaluators scored the flavor intensity of sufu samples, the final result was expressed as the average of the scores of the ten evaluators.

### 3.5. Statistical Analysis

The alpha diversity, bacterial community, and redundancy analysis (RDA) were performed by the online Majorbio Cloud Platform (www.majorbio.com, 29 July 2022). Highly relevant microorganisms and metabolites were visualized using Cytoscape (v.3.7.1) sofeware. Correlations among dominant genera were visualized using Gephi (v.0.9.2). Heatmap analysis between core microbiota and environmental factors was performed using TBtools 0.665. A principal component analysis (PCA) model was established to examine differences among the sufu samples using the SIMCA-14.1 software package. The ANOVA analysis was performed by SPSS 25.0. The read sequences obtained from Illumina MiSeq were submitted to the NCBI database (accession numbers: SRR19152823 to SRR19152843).

## 4. Conclusions

In this study, the fermentation process of sufu was divided into the early and late stages. The early stage is the key period for flavor formation, whereas flavor remained stable due to constant microbial diversity in the late stage. This study investigated relationships among bacteria, characteristic flavor compounds, and physicochemical factors based on each stage. Five bacterial genera, namely *Pseudomonas*, *Tetragenococcus*, *Lysinibacillus*, *Pantoea*, and *Burkholderia*–*Caballeronia*–*Paraburkholderia*, were identified as core microbiota in the early stage, whereas three bacterial genera (*Exiguobacterium*, *Bacillus*, and *Pseudomonas*) were identified as core microbiota in the late stage. The findings of KEGG analysis indicated that the seven microbial genera mainly affect flavor formation during sufu fermentation. The synergistic and exclusive relationships between them maintained the stability of sufu fermentation. Salt and ethanol content were the key physicochemical factors affecting the growth of core microbiota. These results provide new insights for improving sufu flavor to produce high-quality fermented foods.

## Figures and Tables

**Figure 1 molecules-27-04933-f001:**
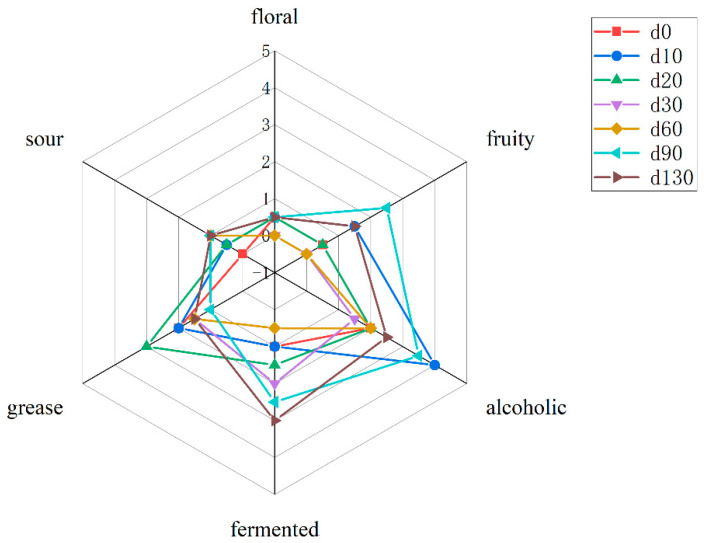
The radar plot of flavor characteristics during sufu fermentation. A score of 0 represents non-existent, 1 represents very slight, 2 represents slight, 3 represents obvious, 4 represents significant, and 5 represents highly significant.

**Figure 2 molecules-27-04933-f002:**
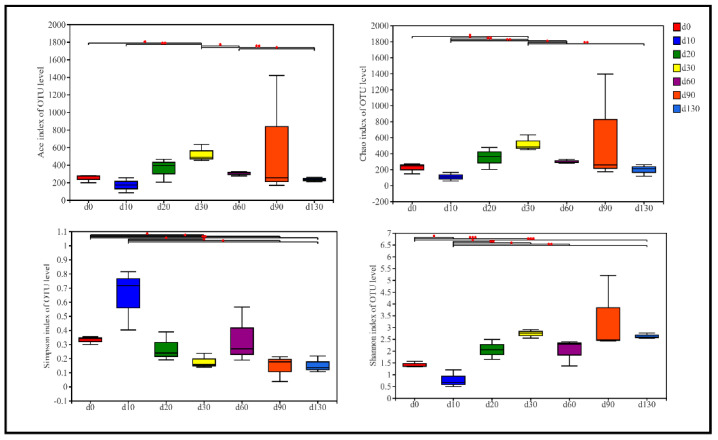
The dynamics of bacterial community aphla diversity indices including chao, ace, Simpson and Shannon during sufu fermentation, significant value are shown as: * *p* < 0.05, ** *p* < 0.01, *** *p* < 0.001.

**Figure 3 molecules-27-04933-f003:**
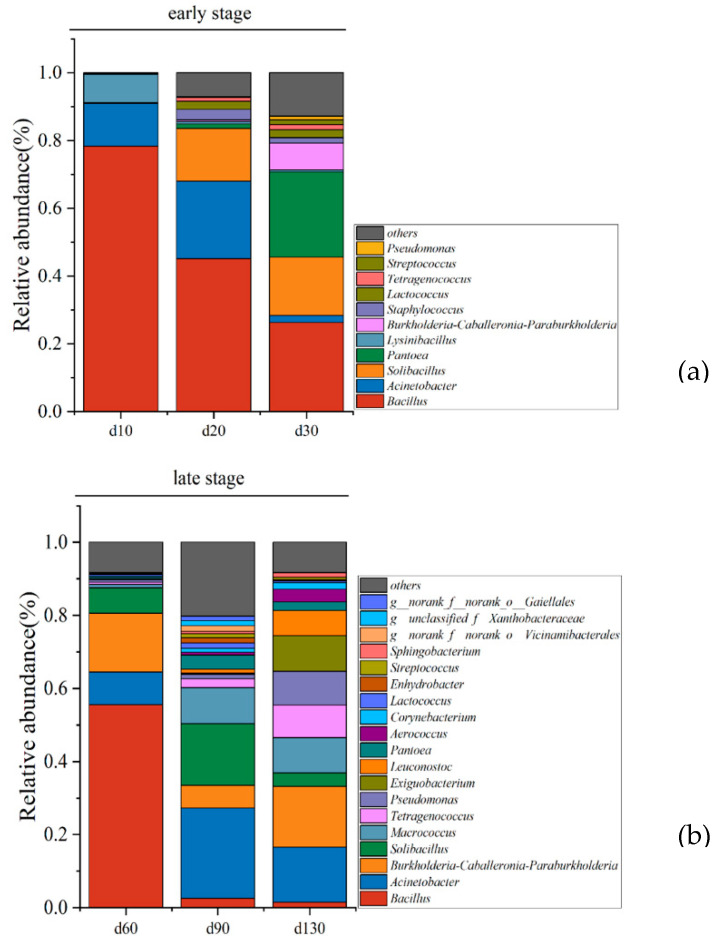
Bacterial community succession at the genus level in the early stage (**a**) and in the late stage (**b**).

**Figure 4 molecules-27-04933-f004:**
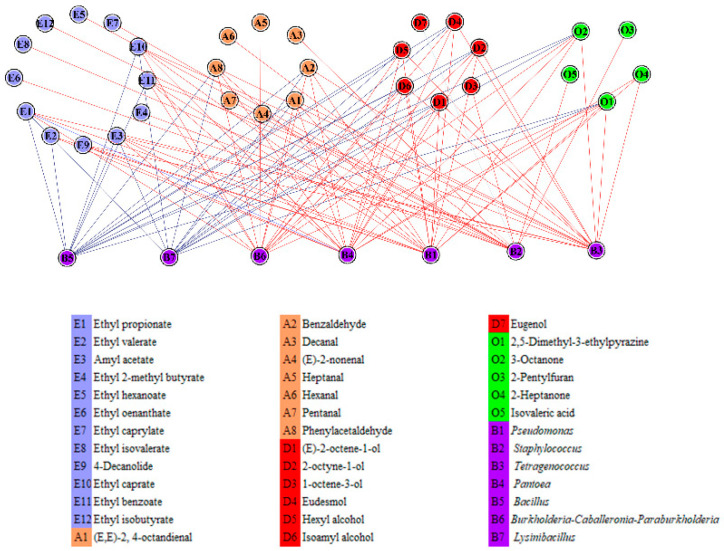
Correlation network analysis of characteristic flavor components and dominant genera in the early stage of sufu fermentation, the red line represents a positive correlation and the blue line represents a negative correlation.

**Figure 5 molecules-27-04933-f005:**
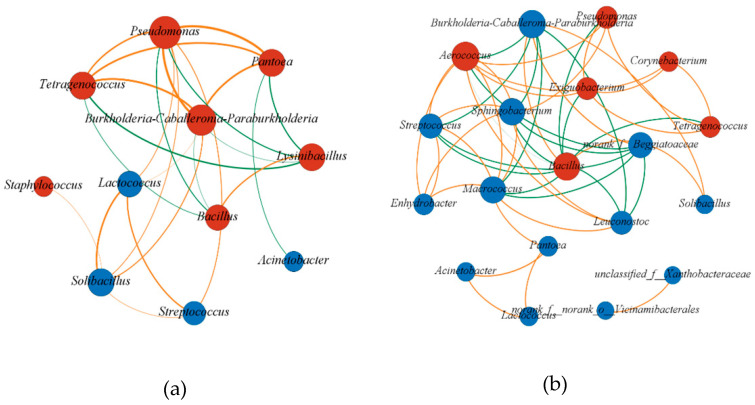
Co-occurrence network analysis of dominant genera in the early stage (**a**) and in the late stage (**b**), the red nodes are strongly flavor-related genus and the blue nodes are other dominant bacteria, the yellow side shows a positive correlation while the green side shows a negative correlation.

**Figure 6 molecules-27-04933-f006:**
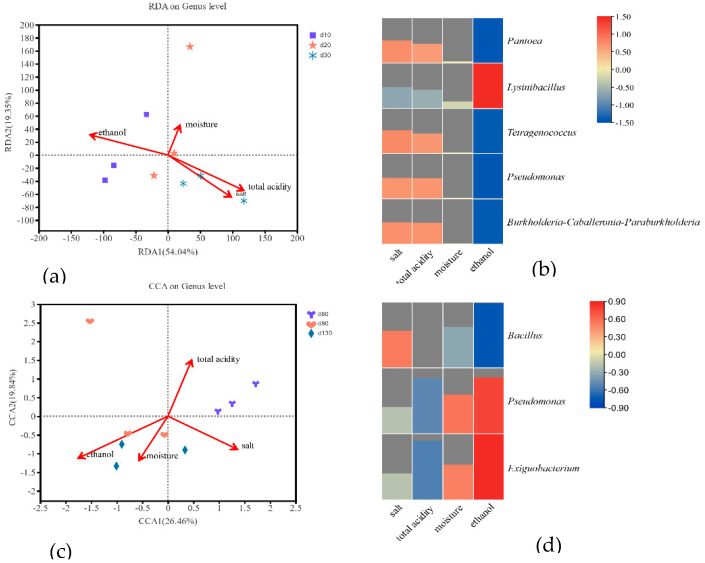
The redundancy analysis for evaluating the distributions of physicochemical factors to bacterial community in the early stage (**a**) and in the late stage (**c**), heatmap analysis of core microbiota with physicochemical factors in the early stage (**b)** and in the late stage (**d**), where the peak height represents the magnitude of the correlation.

**Figure 7 molecules-27-04933-f007:**
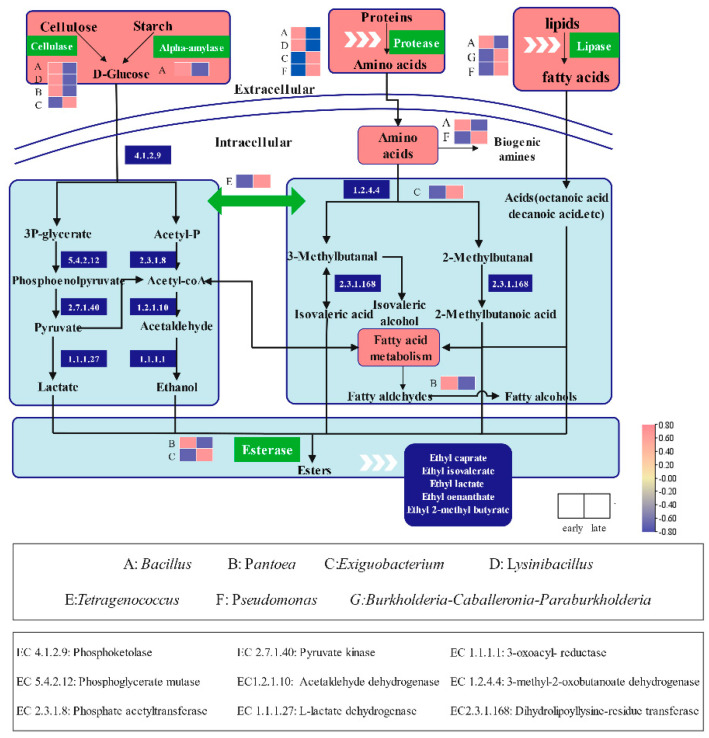
The key metabolic pathways involved by seven core microbiota of aroma compounds formation in sufu fermentation, the green blocks represent Involved enzymes, the red blocks in the heat map represent the early stage, and the blue blocks represent the late stage.

**Table 1 molecules-27-04933-t001:** Characteristic volatile flavor compounds of sufu samples.

Volatile Flavor Compounds	Threshold (ug/g) ^b^	Description ^c^	Sufu Samples (ug/100 g) ^a^						
			d0	d10	d20	d30	d60	d90	d130
Esters (14)									
Ethyl propionate	0.01	Pineapple	−	−	4.28 ± 0.68 ^d^	5.63 ± 0.56 ^c^	6.02 ± 1.08 ^c^	15.72 ± 0.54 ^a^	7.73 ± 1.22 ^b^
Ethyl butanoate	0.001	Fruity, banana	28.2 ± 0.32 ^a^	22.26 ± 0.51 ^b^	27.43 ± 3.17 ^a^	20.13 ± 0.36 ^b^	21.01 ± 0.58 ^b^	16.64 ± 0.34 ^c^	16.49 ± 0.25 ^c^
Ethyl isobutyrate	0.0001	Fruity	−	2.61 ± 1.13 ^c^	9.62 ± 0.74 ^a^	6.30 ± 1.01 ^b^	9.69 ± 1.96 ^a^	10.09 ± 0.91 ^a^	9.65 ± 0.52 ^a^
Ethyl 2-methyl butyrate	0.00015	Grassy	5.71 ± 1.36 ^e^	20.52 ± 0.26 ^d^	54.25 ± 4.62 ^a^	33.60 ± 1.47 ^b^	18.06 ± 1.35 ^d^	30.45 ± 1.01 ^b^	26.65 ± 0.25 ^c^
Ethyl valerate	0.0058	Fruity	27.30 ± 0.56 ^c^	29.30 ± 2.34 ^c^	36.53 ± 4.32 ^b^	36.05 ± 2.86 ^b^	44.40 ± 3.19 ^a^	27.75 ± 1.59 ^c^	42.49 ± 3.33 ^a^
Ethyl isovalerate	0.0002	Nail polish	−	6.02 ± 0.16 ^e^	15.65 ± 1.58 ^a^	9.56 ± 0.39 ^c^	5.88 ± 0.32 ^e^	8.13 ± 0.34 ^d^	13.64 ± 0.22 ^b^
Emyl acetate	0.05	−	6.32 ± 0.49 ^c^	5.69 ± 0.95 ^c^	7.00 ± 0.50 ^bc^	11.41 ± 0.65 ^a^	11.8 ± 1.56 ^a^	8.17 ± 2.05 ^b^	6.39 ± 0.89 ^c^
Ethyl hexanoate	0.005	Almond, apple	217.70 ± 14.92 ^f^	264.25 ± 13.47 ^e^	430.11 ± 17.47 ^c^	339.45 ± 21.64 ^d^	465.58 ± 13.39 ^b^	321.63 ± 21.61 ^d^	618.33 ± 24.81 ^a^
Ethyl oenanthate	0.0019	Fruity	101.95 ± 7.33 ^d^	105.45 ± 7.40 ^d^	181.42 ± 4.21 ^b^	149.67 ± 13.81 ^c^	156.23 ± 4.58 ^c^	106.57 ± 3.42 ^d^	216.38 ± 9.13 ^a^
Ethyl caprylate	0.0193	Creamy	−	160.78 ± 14.78 ^c^	252.49 ± 3.37 ^b^	230.12 ± 26.79 ^b^	326.10 ± 20.21 ^a^	249.80 ± 14.68 ^b^	329.22 ± 11.83 ^a^
4-decanolide	0.0026	Fruity, peach	5.61 ± 1.17 ^c^	6.30 ± 1.15 ^bc^	10.97 ± 0.70 ^a^	11.01 ± 0.23 ^a^	7.34 ± 1.59 ^b^	5.57 ± 0.78 ^c^	9.53 ± 0.57 ^a^
Ethyl caprate	0.023	Flower	28.71 ± 1.22 ^b^	11.40 ± 2.39 ^b^	20.80 ± 2.06 ^b^	20.51 ± 0.97 ^b^	16.38 ± 0.87 ^b^	15.62 ± 2.63 ^b^	90.29 ± 10.91 ^a^
Ethyl benzoate	0.053	−	−	−	−	4.52 ± 0.81 ^b^	4.79 ± 0.3 ^b^	4.38 ± 0.80 ^c^	6.64 ± 0.42 ^a^
Propyl(E)-2-methyl-2-Butenoate	0.012	−	−	−	−	−	2.65 ± 1.05 ^b^	1.50 ± 0.12 ^c^	3.90 ± 0.46 ^a^
Alcohols (8)									
Isoamyl alcohol	0.22	Alcoholic	80.14 ± 8.41 ^c^	82.81 ± 6.64 ^c^	93.62 ± 5.67 ^b^	116.33 ± 10.12 ^a^	39.16 ± 8.34 ^d^	40.74 ± 5.44 ^d^	44.30 ± 2.15 ^d^
Hexyl alcohol	0.5	Sour, pungent	85.94 ± 5.05 ^bc^	87.84 ± 5.42 ^bc^	101.42 ± 9.60 ^a^	100.37 ± 3.52 ^a^	96.86 ± 9.22 ^ab^	58.28 ± 3.59 ^d^	79.68 ± 2.98 ^c^
1-octene-3-ol	0.007	Mushroom fragrance	−	106.57 ± 9.17 ^e^	224.22 ± 8.62 ^a^	178.42 ± 19.68 ^d^	201.00 ± 14.91 ^bc^	207.58 ± 1.76 ^ab^	182.19 ± 8.23 ^cd^
2-octyne-1-ol	0.003	−	24.59 ± 1.70 ^c^	20.32 ± 2.78 ^c^	24.86 ± 0.29 ^c^	57.40 ± 6.50 ^a^	31.40 ± 2.88 ^b^	20.55 ± 0.40 ^c^	13.09 ± 0.86 ^d^
(E)-2-octene-1-ol	0.02	−	9.20 ± 0.65 ^c^	7.88 ± 0.24 ^c^	14.18 ± 2.02 ^b^	41.97 ± 3.54 ^a^	10.74 ± 1.42 ^c^	7.93 ± 0.16 ^c^	8.78 ± 0.42 ^c^
Eudesmol	0.003	Herb	−	5.28 ± 1.72 ^d^	4.41 ± 0.06 ^d^	26.14 ± 1.73 ^a^	13.60 ± 2.72 ^b^	8.19 ± 0.14 ^c^	10.67 ± 0.82 ^c^
Linalool	0.006	Lily	4.48 ± 1.26 ^c^	7.33 ± 1.24 ^b^	5.61 ± 1.09 ^c^	44.75 ± 1.13 ^a^	3.99 ± 0.41 ^c^	5.51 ± 0.38 ^c^	8.22 ± 0.70 ^b^
1-nonanol	0.05	−	9.22 ± 1.14 ^b^	8.56 ± 2.44 ^bc^	12.28 ± 1.10 ^a^	9.27 ± 1.11 ^b^	6.74 ± 1.30 ^cd^	5.83 ± 0.37 ^d^	5.69 ± 0.25 ^d^
Aldehydes (11)									
Pentanal	0.012	Pungent	14.59 ± 1.75 ^bcd^	13.75 ± 1.61 ^cd^	18.23 ± 0.83 ^abc^	13.02 ± 2.76 ^d^	19.69 ± 0.85 ^a^	14.56 ± 2.43 ^bcd^	18.73 ± 4.68 ^ab^
Hexanal	0.0039	Beany, grassy	55.19 ± 1.60 ^d^	39.57 ± 0.27 ^e^	81.22 ± 4.08 ^c^	59.99 ± 1.68 ^d^	153.63 ± 6.59 ^a^	106.16 ± 6.03 ^b^	56.73 ± 2.82 ^d^
Heptanal	0.0028	Tallow	7.29 ± 0.86 ^cd^	7.32 ± 0.16 ^cd^	8.50 ± 1.48 ^bc^	6.60 ± 1.06 ^d^	12.16 ± 0.81 ^a^	9.05 ± 0.58 ^b^	7.78 ± 0.24 ^bcd^
Octanal	0.0008	Citrus-like	9.90 ± 1.53 ^b^	12.49 ± 2.16 ^b^	10.32 ± 1.39 ^b^	18.22 ± 2.15 ^a^	16.03 ± 2.53 ^a^	9.29 ± 0.57 ^b^	10.09 ± 0.80 ^b^
Benzaldehyde	0.35	Almond	−	−	36.86 ± 0.57 ^b^	38.77 ± 0.57 ^b^	58.36 ± 3.95 ^a^	36.27 ± 2.11 ^b^	38.91 ± 2.63 ^b^
Phenylacetaldehyde	0.004	Honey	14.17 ± 0.23 ^cd^	13.36 ± 2.22 ^cd^	15.95 ± 2.31 ^bc^	43.37 ± 0.37 ^a^	18.73 ± 2.77 ^b^	13.03 ± 0.94 ^cd^	11.06 ± 0.14 ^d^
Nonanal	0.015	Flower, orange	50.85 ± 1.66 ^b^	49.46 ± 1.01 ^b^	28.96 ± 2.12 ^c^	82.06 ± 1.78 ^a^	43.98 ± 9.73 ^b^	23.28 ± 0.42 ^c^	28.55 ± 4.07 ^c^
(E)-2-nonenal	0.00019	Fatty, tallow	9.20 ± 0.65 ^b^	10.48 ± 0.27 ^a^	7.32 ± 0.48 ^d^	9.04 ± 0.60 ^b^	8.64 ± 0.86 ^b^	6.39 ± 0.22 ^e^	5.05 ± 0.14 ^f^
Decanal	0.005	Fatty	12.86 ± 0.82 ^c^	24.20 ± 0.62 ^a^	15.24 ± 1.96 ^c^	20.42 ± 1.11 ^ab^	20.23 ± 5.17 ^ab^	14.91 ± 1.42 ^c^	17.04 ± 2.13 ^bc^
(E,E)-2, 4-octandienal	0.00001	−	−	−	10.11 ± 0.98 ^a^	18.99 ± 0.45 ^b^	−	−	−
2,4-undecadienal	0.00001	−	8.29 ± 0.90 ^a^	4.23 ± 0.64 ^b^	8.14 ± 0.29 ^a^	4.42 ± 0.74 ^b^	5.20 ± 0.19 ^b^	5.45 ± 1.26 ^b^	−
Ketones (3)									
3-octanone	0.057	−	11.38 ± 0.84 ^e^	17.03 ± 1.12 ^d^	30.45 ± 2.61 ^a^	26.35 ± 1.63 ^b^	24.06 ± 0.57 ^bc^	23.21 ± 0.20 ^c^	18.03 ± 0.75 ^d^
2-nonanone	0.05	Coconut-like	8.67 ± 0.08 ^b^	7.50 ± 1.30 ^bc^	6.02 ± 1.94 ^cd^	39.41 ± 2.72 ^a^	4.22 ± 0.46 ^d^	4.35 ± 0.26 ^d^	6.66 ± 0.03 ^bcd^
2-heptanone	0.14	−	7.80 ± 0.76 ^e^	10.83 ± 0.49 ^d^	11.96 ± 1.78 ^cd^	13.75 ± 1.28 ^c^	16.74 ± 1.44 ^b^	12.60 ± 0.51 ^cd^	22.69 ± 0.07 ^a^
Phenols (2)									
Eugenol	0.0071	Clove	−	23.64 ± 3.26 ^d^	39.99 ± 3.68 ^b^	28.84 ± 0.12 ^c^	28.05 ± 1.76 ^c^	27.28 ± 1.83 ^cd^	52.67 ± 1.69 ^a^
4-ethenyl-2-Methoxyphenol	0.00001	−	10.22 ± 0.97 ^d^	16.01 ± 1.98 ^c^	11.74 ± 0.84 ^d^	31.36 ± 0.04 ^a^	21.06 ± 1.45 ^b^	17.24 ± 0.33 ^c^	29.35 ± 3.17 ^a^
Others (3)									
Isovaleric acid	0.1	Acid, rancid	−	12.90 ± 5.02 ^c^	10.47 ± 0.54 ^a^	9.60 ± 1.15 ^a^	5.79 ± 2.05 ^b^	4.77 ± 0.93 ^b^	−
2-pentylfuran	0.0058	Beany	120.44 ± 8.32 ^ab^	75.61 ± 7.34 ^d^	126.84 ± 9.71 ^a^	112.52 ± 6.23 ^b^	93.25 ± 7.17 ^c^	98.25 ± 7.82 ^c^	87.05 ± 4.43 ^cd^
2,5-Dimethyl-3-Ethylpyrazine	0.0086	Roast	−	7.14 ± 1.31 ^c^	13.552 ± 0.56 ^b^	36.54 ± 1.46 ^a^	2.91 ± 1.58 ^d^	4.16 ± 0.12 ^d^	−

^a^ This value were the mean values of three parallel samples, “−” represented the compound was not detected, means with different superscript letters are significantly different horizontally(one-way analysis of variance; *p* < 0.05). ^b^ The thresholds for flavor compounds were detected in water media, with reference to [21]. ^c^ The aroma description of flavor compounds were obtained from [10]. ^d^ Changes in flavor properties of sufu during fermentation.

## Data Availability

The read sequences obtained from Illumina MiSeq are available from the NCBI database. The other data that supports the findings of this study are available in the Appendix A of this article.

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
