# Peer review of "Exploring Core Microbiota Based on Characteristic Flavor Compounds in Different Fermentation Phases of Sufu"

_molecules, 2022, doi:10.3390/molecules27154933_

Round 1

Reviewer 1 Report

The manuscript of Wu Wei et al is devoted to the study of the formation of characteristic flavor components at various stages of sufu fermentation. The work is written in a competent scientific language. The results and applied approaches are of scientific interest and can be used to control production processes and the quality of food products. I believe that the manuscript can be accepted for publication in the journal "Molecules" in its present form.

The work is aimed at characterizing the microbiota, which has an important contribution to the formation of flavor compounds in the production of sufu. The study addresses the issue of identifying the most important types of microbiota and the mechanisms of their metabolism that occur during the fermentation of a soybean-based product.

The topic is quite interesting. The approaches applied by the authors can be successfully used to improve the technology of food products, to impart the necessary taste properties. In addition, the control of flavor compounds formed during fermentation processes is an important part of improving the quality of the finished product.

The study revealed not only the most important representatives of the microbiota and their relationships with each other, but also their contribution to the formation of flavor compounds.

I did not notice any problems with the methodology used by the authors. In my opinion, it would not be bad to study not only volatile compounds, but also non-volatile ones. Perhaps some important taste components are not detected. The use of HPLC with high resolution mass spectrometry could potentially greatly expand the range of substances that play a role in the formation of flavor during fermentation.

Table 1 in the manuscript is largely duplicated with a table in supplementary materials. I think the authors should leave only 1 option.

Author Response

Point 1: I did not notice any problems with the methodology used by the authors. In my opinion, it would not be bad to study not only volatile compounds, but also non-volatile ones. Perhaps some important taste components are not detected. The use of HPLC with high resolution mass spectrometry could potentially greatly expand the range of substances that play a role in the formation of flavor during fermentation.

Response 1: Thank you for your encouragement and valuable comment, non-volatile flavor compounds are also worth studying, and this is the direction of our future research.

Point 2: Table 1 in the manuscript is largely duplicated with a table in supplementary materials. I think the authors should leave only 1 option.

Response 2: Thanks for the valuable suggestion. As suggested, we have deleted Figure S1 from supplementary materials.

Reviewer 2 Report

Please state the statistical analysis in the materials and method section, which software was used for ANOVA analysis?

Table 1, no statistical analysis?

The discussion can be improved by comparing with previous studies. 

Author Response

Point 1: Please state the statistical analysis in the materials and method section, which software was used for ANOVA analysis?

Response 1: Thank you for your helpful suggestion. The SPSS 25.0 was used for ANOVA analysis, and we have added the software version of ANOVA analysis to the statistical analysis section(shown in Lines 153-154).

Point 2: Table 1, no statistical analysis?

Response 2: Thanks for your helpful suggestion, We have performed an ANOVA analysis on the data in Table 1 and marked the data with different letters in the upper right corner, different letters indicates different degrees of variation.

Point 3: The discussion can be improved by comparing with previous studies.

Thanks for your helpful suggestion, the discussion of microbial-flavor correlation analysis focuses on section 3.6, as suggested, we have added comparative analysis with previous studies(shown in Lines 349-351),and improved the comparative analysis with previous studies(shown in Lines 337-341).

Reviewer 3 Report

There is an interesting manuscript, however, I have some suggestions for the authors

1. In the part of the introduction,  the authors can write more information about Sufu.

2. In the experiment, the authors can do a sensory evaluation for the research.

Thanks.

Author Response

Point 1: In the part of the introduction, the authors can write more information about Sufu.

Response 1: Thank you for your helpful suggestion. We have added more information about sufu in the introduction section(shown in Lines 30-33).

Point 2: In the experiment, the authors can do a sensory evaluation for the research.

Response 2: Thanks for your helpful suggestion. Sensory evaluation experiments on the flavor of sufu samples have been completed previously, as suggested, we have added the sensory evaluation of sufu to the manuscript (shown in Lines 134-145 and Lines 210-222).